# Antibiotics can be used to contain drug-resistant bacteria by maintaining sufficiently large sensitive populations

Elsa Hansen[1]☉, Jason Karslake[2]☉, Robert J. Woods[3], Andrew F. Read[4], Kevin B. Wood[2,5]*

**1** Center for Infectious Disease Dynamics, Department of Biology, Pennsylvania State University, University Park, Pennsylvania, United States of America, **2** Department of Biophysics, University of Michigan, Ann Arbor, Michigan, United States of America, **3** Division of Infectious Diseases, Department of Internal Medicine, University of Michigan, Ann Arbor, Michigan, United States of America, **4** Center for Infectious Disease Dynamics, Huck Institutes of the Life Sciences and Departments of Biology and Entomology, Pennsylvania State University, University Park, Pennsylvania, United States of America, **5** Department of Physics, University of Michigan, Ann Arbor, Michigan, United States of America

☉ These authors contributed equally to this work.
* kbwood@umich.edu

**Data Availability Statement:** Data deposited in the Dryad repository: https://doi.org/10.5061/dryad.s4mw6m943.

## Abstract

Standard infectious disease practice calls for aggressive drug treatment that rapidly eliminates the pathogen population before resistance can emerge. When resistance is absent, this elimination strategy can lead to complete cure. However, when resistance is already present, removing drug-sensitive cells as quickly as possible removes competitive barriers that may slow the growth of resistant cells. In contrast to the elimination strategy, a containment strategy aims to maintain the maximum tolerable number of pathogens, exploiting competitive suppression to achieve chronic control. Here, we combine in vitro experiments in computer-controlled bioreactors with mathematical modeling to investigate whether containment strategies can delay failure of antibiotic treatment regimens. To do so, we measured the "escape time" required for drug-resistant *Escherichia coli* populations to eclipse a threshold density maintained by adaptive antibiotic dosing. Populations containing only resistant cells rapidly escape the threshold density, but we found that matched resistant populations that also contain the maximum possible number of sensitive cells could be contained for significantly longer. The increase in escape time occurs only when the threshold density—the acceptable bacterial burden—is sufficiently high, an effect that mathematical models attribute to increased competition. The findings provide decisive experimental confirmation that maintaining the maximum number of sensitive cells can be used to contain resistance when the size of the population is sufficiently large.

## Introduction

The ability to successfully treat infectious disease is often undermined by drug resistance [1–6]. When resistance poses a major threat to the quality and duration of a patient's life, the goal

**Funding:** This work is supported by the National Science Foundation (NSF No. 1553028 to KBW), the National Institutes of Health (NIH No. 1R35GM124875-01 to KBW; NIH No. R01 GM089932 to AFR; NIH K08 AI119182 to RJW), the Hartwell Foundation for Biomedical Research (to KBW), and the Eberly Family (to AFR). The funders had no role in study design, data collection and analysis, decision to publish, or preparation of the manuscript.

**Competing interests:** The authors have declared that no competing interests exist.

**Abbreviations:** LTEE, long-term evolution experiment; OD, optical density; $P_{max}$, acceptable burden; TA, tetrazolium arabinose.

of treatment is to restore patient health while delaying treatment failure for as long as possible. To do so, standard practice calls for aggressive drug treatment to rapidly remove the drug-sensitive pathogen population and prevent resistance-conferring mutations [7–17]. Aggressive treatment can involve either single-drug or combination therapies, which have been shown to modulate the emergence of resistance [18–25]. Here, we are interested in situations in which such aggressive regimens do not completely prevent the emergence of resistance—for example, scenarios in which resistance is already present at the onset of treatment.

If aggressive treatment cannot prevent the emergence of resistance, an alternative approach is to use competition between drug-sensitive and drug-resistant cells to slow the expansion of the drug-resistant population. There is ample evidence that competition between sensitive and resistant cells can be intense [26–29] and may be over limited resources like glucose or target cells [30–33]. Competition can also be immune mediated or occur via direct interference (e.g., bacteriocins) [26, 34–37]. There are numerous theoretical studies [35, 38–49] suggesting that sensitive cells can competitively suppress resistant cells, and this suppression has even been observed experimentally in parasites and cancer [42, 50–55]. Ideally, resistance never emerges, but if it does, delaying the time to treatment failure can potentially prolong life (chronic infections [56]) or give immunity time to prevent resistance emergence (e.g., acute infections, or when immunosuppression is medically induced and temporary). Because sensitive cells can both generate de novo resistance and also competitively suppress existing resistant mutants, making good treatment decisions requires understanding the relative importance of these opposing effects (Fig 1).

Recent theoretical work compares two extreme treatment strategies: a strategy that removes all drug-sensitive cells (what we call elimination) and a strategy that maximizes the sensitive population (what we call containment) [45]. The elimination strategy removes sensitive cells as fast as possible, minimizing the risk of mutation but also removing competitive barriers that may slow the growth of existing resistant cells [35] (Fig 1A). Containment, on the other hand, maintains as many sensitive pathogens as is clinically acceptable (i.e., a pathogen density that is deemed to be safe and below which treatment is not necessary), using drug treatment only to alleviate symptoms. Containment maintains the total pathogen density at this acceptable burden (Fig 1B). This maximizes competitive suppression but leaves sensitive pathogens that

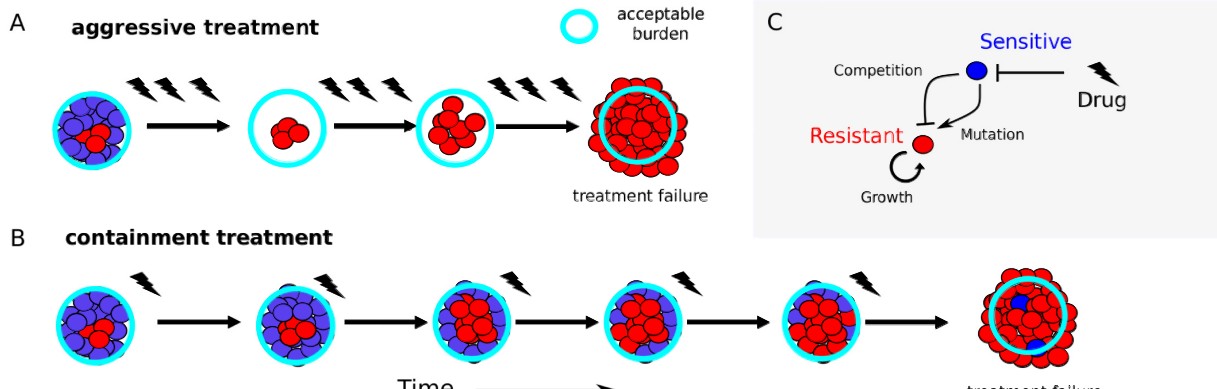

**Fig 1. Containment strategies may leverage competition to extend time below treatment failure threshold.** (A) Aggressive treatment uses high drug concentrations (lightning flashes), which eliminates sensitive cells (blue) but may fail when resistant cells (red) emerge and the population exceeds the failure threshold ("acceptable burden", light-blue circle). (B) Containment strategies attempt to maintain the population just below the failure threshold, leveraging competition between sensitive (blue) and emergent resistant (red) cells to potentially prolong time to failure. (C) Schematic of potential feedback between growth processes in mixed populations. Drug (lightning flash) inhibits sensitive cells (blue), which in turn inhibit resistant cells (red) through competition but may also contribute to the resistant population via mutation.

can generate resistance. Cases in which containment better slows the expansion of the resistant population represent situations in which standard practice can potentially be improved.

Theory predicts that neither strategy is always best; there are situations in which containment will better control resistance and others in which it will make a bad prognosis worse [45]. The latter case arises when the benefit of maintaining sensitive pathogens (competitive suppression of resistance) is outweighed by the rate at which sensitive pathogens become resistant by mutation (or horizontal gene transfer). Mathematical modeling suggests that across a wide variety of diseases and settings, the same fundamental principles determine when containment is better than elimination. First, there has to be sufficient competition (the clinically acceptable burden high enough), and second, the growth of the resistant population has to be driven primarily by the replication of resistant cells, not by mutational inputs (resistant population large enough).

Here, we experimentally test the hypothesis that containment strategies that maintain a subpopulation of sensitive cells can slow the expansion of resistant bacteria. We combine simple mathematical models with in vitro experiments in computer-controlled bioreactors, in which "treatment failure" is defined by bacterial populations eclipsing a threshold density (the acceptable burden). The experimental design directly tests the effect of maximizing the size of the sensitive subpopulation on escape time. We find that containment strategies can increase escape times when the acceptable burden is sufficiently high but are ineffective at low densities, in which competition is small. Although there is empirical evidence for competitive suppression in parasites and cancer [50–55], our work provides an explicit demonstration of competitive suppression of resistance to antibiotic treatment and a direct test of a competition-maximizing containment strategy in a bacterial pathogen. The findings are particularly striking because they occur in well-mixed populations with a continual renewal of resources and using an acceptable burden well below the natural carrying capacity—all conditions not typically associated with strong competition.

## Results

The aim of this study is to investigate whether a containment strategy that maintains subpopulations of sensitive cells can improve our ability to keep drug-resistant populations below a predefined threshold density. To do so, we develop an experimental assay based on adaptive drug dosing that allows us to directly compare escape times for resistant-only populations with those of matched resistant populations supplemented with sensitive populations of different sizes.

### Model system

We grew bacterial populations in well-mixed bioreactors in which environmental conditions, including drug concentration and nutrient levels, can be modulated using a series of computer-controlled peristaltic pumps (see, for example, [57, 58]). Population density is measured using light scattering (optical density [OD]), and drug concentration can be adjusted in real time in response to population dynamics or predetermined protocols (S1 Fig).

To obtain bacterial populations with different drug sensitivities, we began with *E. coli* strains REL606 and REL607, which are well-characterized ancestral strains used in the long-term evolution experiment (LTEE) in *E. coli* [59]. These strains differ by a single point mutation in *araA*, which serves as a neutral marker for competition experiments; REL606 (REL607) appears red (pink) when grown on tetrazolium arabinose (TA) plates. To generate a "drug-resistant" strain, we used laboratory evolution to isolate a mutant of the REL606 strain that was resistant to doxycycline, a frequently used protein synthesis inhibitor (Methods). Whole-genome sequencing confirmed that the ancestor REL606 strain was identical to the published sequence (accession number NC_012967), whereas the REL606-derived resistant strain had

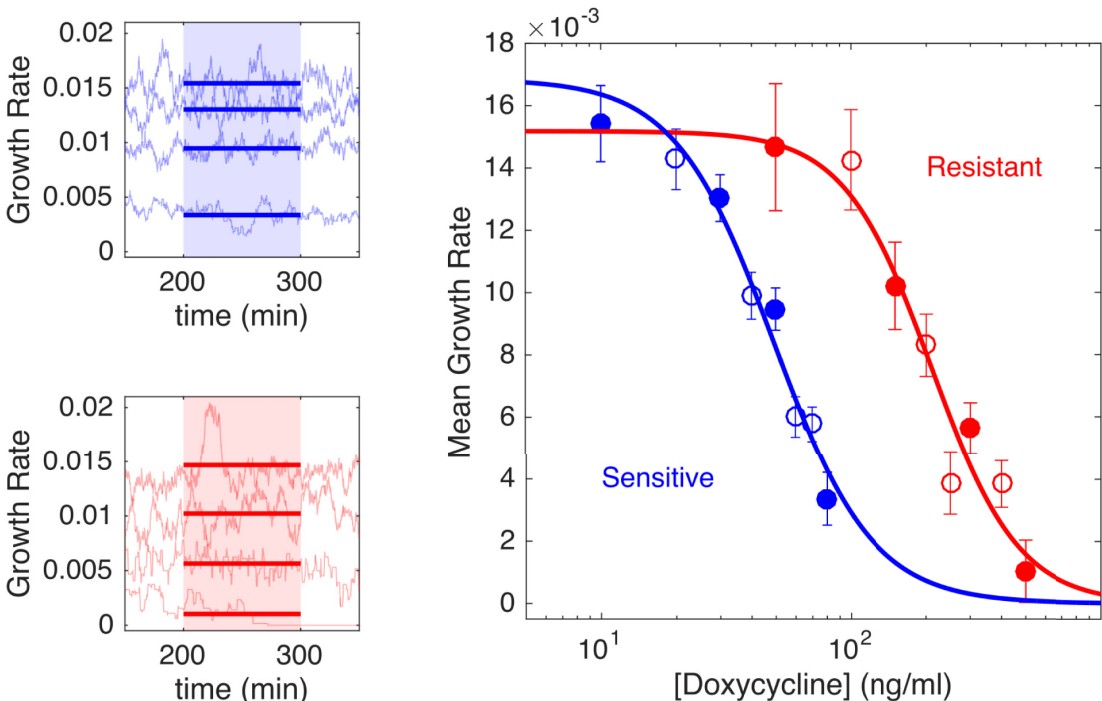

**Fig 2. Resistant cells exhibit increased resistance to doxycycline and a small fitness cost.** Left panels: per capita growth rate in bioreactors for ancestral (sensitive, blue) and resistant (red) populations exposed to increasing concentrations of doxycycline (top to bottom in each panel). Real-time per capita growth rate (light blue or red curves) is estimated from flow rates required to maintain constant cell density at each drug concentration (Methods). Mean growth rate (thick solid lines) is estimated between 200 and 300 minutes post–drug addition (shaded regions), when the system has reached steady state. Doxycycline concentrations are 10, 30, 50, and 80 ng/mL (top panel, top to bottom) and 50, 150, 300, and 500 ng/mL (bottom panel, top to bottom). Right panel: dose-response curve for sensitive (blue) and resistant (red) populations. Circles are time-averaged growth rates between 200 and 300 minutes post–drug addition (shaded regions in [A]), with error bars ± one standard deviation over the measured interval; filled circles correspond to the specific examples shown in left panels. Drug-free growth rates are 0.017±0.001 minutes$^{-1}$ (sensitive) and 0.015±0.002 minutes$^{-1}$ (resistant), indicating that resistant cells exhibit a fitness cost. Solid lines, fit to Hill-like dose-response function $r = r_0(1+(D/h)^k)^{-1}$, with $r$ the growth rate, $D$ the drug concentration, $r_0$ the growth in the absence of drug, $h$ the IC$_{50}$, and $k$ the Hill coefficient. IC$_{50}$ values are estimated to be $h = 49$ ng/mL (sensitive cells) and $h = 210$ ng/mL (resistant cells). Data are deposited in the Dryad repository: https://doi.org/10.5061/dryad.s4mw6m943 [62]. IC$_{50}$, half-maximal inhibitory concentration.

five mutations, including mutations in *OmpF* and *AcrR*, genes that have been previously linked with increased resistance to multiple antibiotics, including tetracyclines [60, 61]. We quantified the phenotypic responses of the drug-sensitive (REL607) and drug-resistant (REL606-derived mutant) cells to doxycycline by measuring real-time per capita growth rate for isogenic populations of each strain exposed to different concentrations of drug (Fig 2). Briefly, growth rate was estimated using influx rate of media required to maintain populations at a constant density (Methods). The resistant isolate exhibits both increased resistance to doxycycline (increased half-maximal inhibitory concentration) as well as decreased growth in the absence of drug (Fig 2). We note that in the experiments that follow, drug concentrations are sufficiently high such that resistant cells generally have a selective advantage over sensitive cells despite this fitness cost.

## Experimental design to measure effect of sensitive cells on escape time

Our experimental design aims to measure the effect of sensitive cells on a resistant population. First, we seed two vials with the same low density of resistant cells. Then, drug-sensitive cells

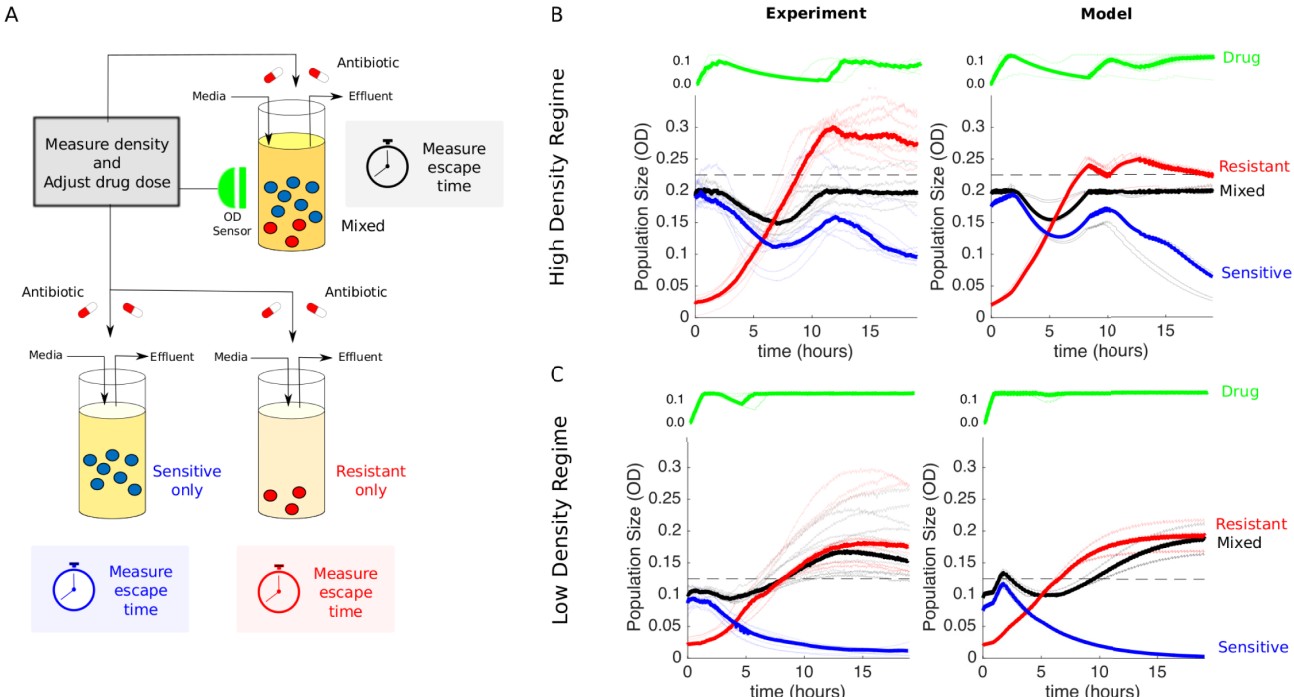

**Fig 3. Experimental design and dynamics for comparison to measure effect of "competition maximization".** (A) Schematic of experiment. Three different populations (sensitive-only, resistant-only, and mixed) were exposed to identical antibiotic treatments in separate bioreactors. The medium in each bioreactor was also refreshed at a constant rate of $F_N = 0.067$ mL/min. The drug treatment was determined in real time by measuring the density (OD) of the mixed population and adjusting drug influx to maintain a constant density ($P_{max}$) while minimizing drug used (Methods). Although the dynamics of the mixed population fully determine the temporal profile of the drug dosing, all three populations then receive identical treatments. In the high-density experiment, mixed populations started at an OD of $P_{max} = 0.2$, with a 90–10 ratio of sensitive to resistant cells. The initial OD of resistant cells is therefore 0.02. Resistant-only populations started from an initial density of 0.02 and contained no sensitive cells, whereas sensitive-only populations started from an initial density of 0.18 and contained no resistant cells. In the low-density experiment, mixed populations started at an OD of $P_{max} = 0.1$, and the initial OD of resistant cells was unchanged (OD = 0.02). Therefore, the starting conditions of the high- and low-density experiments differ only in the number of sensitive cells. (B and C) Experiments (left) and model (right) in high-density ($P_{max} = 0.2$ [B]) and low-density ($P_{max} = 0.1$ [C]) regimes. Red curves are resistant only, blue are sensitive only, and black are the mixed populations. Lightly shaded curves correspond to individual experiments, and dark curves show the median across experiments. Horizontal dashed lines show the treatment failure threshold $P_{max} = 0.025$, in which the 0.025 term allows for small experimental fluctuations without triggering a threshold crossing event. Data are deposited in the Dryad repository: https://doi.org/10.5061/dryad.s4mw6m943 [62]. OD, optical density; $P_{max}$, acceptable burden.

are added to one of these vials (the "mixed" vial) to achieve a total cell density equal to a predetermined threshold density—the acceptable burden ($P_{max}$). We use an additive design because we are interested in comparing the dynamics of the resistant population in the presence/absence of the sensitive population. Because high concentrations of drug are expected to completely inhibit growth of sensitive cells and therefore eliminate any potential competition, we designed an adaptive drug dosing protocol intended to maintain the mixed population at a fixed density ($P_{max}$) using minimal drug. The dosing protocol uses simple feedback control to adjust the drug concentration in real time in response to changes in population density (Fig 3A and Methods). Finally, as a control, a third vial is seeded with the same density of sensitive bacteria as that added to the mixed vial (Fig 3A, sensitive-only). This control allows for an indirect measure of the effect of sensitive cells in the mixed vial because, in the absence of competition or other intercellular interactions, the dynamics of the mixed population should be a simple sum of the dynamics in the two single-strain populations.

The temporal dynamics of the mixed population—but not the other populations—completely determine the drug dosing protocol, but all three populations receive identical

drug dosing and therefore experience identical drug concentrations over time. This strategy ensures that any increased resistance suppression in the mixed vial can be attributed to competitive suppression by sensitive cells and not drug inhibition. It is essential that all vials have the same drug concentration—otherwise, differences in resistance suppression could be attributed to the differences in drug concentration. Because drug concentration in the vials is restricted to a finite range (0–125 ng/mL), populations containing resistant cells cannot be contained indefinitely and will eventually eclipse the threshold density ($P_{max}$). The time required for this crossover is defined as the escape time, and the goal of the experiment is to compare escape times—which correspond, intuitively, to times of treatment failure—in the absence of sensitive cells and the presence of the largest acceptable sensitive population.

There are three possibilities. If mutation dominates, then the absence of sensitive cells is expected to be best, and the escape time of the resistant-only population should exceed that of the mixed population. On the other hand, if competition dominates, then the mixed population should take the longest to escape. Finally, if the effects of sensitive cells (both mutational input and competitive suppression) are negligible, then the escape times of the resistant-only and mixed populations should be similar. To quantitatively guide our experiments and refine this intuition, we developed and parameterized a simple mathematical model for population growth in the bioreactors (see Methods and S1 Text for a detailed description of the model). A simple analysis suggests that, for our experimental design, the effect of competition will always dominate over the effect of mutational input (see S2 Text). As a result, we neglect mutation and focus on the role of competition. This allows us to identify two values of acceptable burden ($P_{max}$), which are predicted to produce different results. For high $P_{max}$ (OD = 0.2), competition dominates, and the mixed vial should have the longest escape time. For low $P_{max}$ (OD = 0.1), competition is minimal, and the escape times of the resistant-only and mixed vials should be similar.

## Benefit of competition maximization depends on acceptable burden

To test these predictions, we first performed the experiment at the threshold density that the model predicts will lead to competitive suppression ($P_{max}$ = 0.2 [Fig 3B]). Note that this density falls in the range of exponential growth and falls below the stationary phase limit in unperturbed populations (S2 Fig). To account for batch effects and day-to-day experimental fluctuations, we repeated the experiment multiple times across different days, using different media and drug preparations. Unsurprisingly, the experiments confirm that sensitive-only populations are significantly inhibited under this treatment protocol and never reach the containment threshold; in fact, the overall density decreases slowly over time because of a combination of strong drug inhibition and effluent flow (Fig 3B, blue curves). By contrast, the resistant-only population grows steadily and eclipses the threshold in 6–9 hours (Fig 3B, red curves). Remarkably, however, the mixed population (black curves) is contained below threshold—in almost all cases—for the entire length of the experiment, which spans more than 18 hours. At the end of the experiment, we plated representative examples of resistant-only and mixed populations (S4 Fig), which confirmed that the mixed vial was predominantly resistant at the end of the experiment. In addition, these sensitive and resistant isolates exhibited similar dose-response curves to the original sensitive and resistant strains (S5 Fig). Matched drug-free controls indicate that containment in the mixed vial is not due to artifacts from media inflow or outflow (S1 Text and S3 Fig). The experiments also show remarkable agreement with the model (with no adjustable parameters; compare left and right panels in Fig 3B).

If competition were driving the increased escape time, one would expect the effect to be reduced as the threshold density ($P_{max}$) is decreased. To test this hypothesis, we repeated the

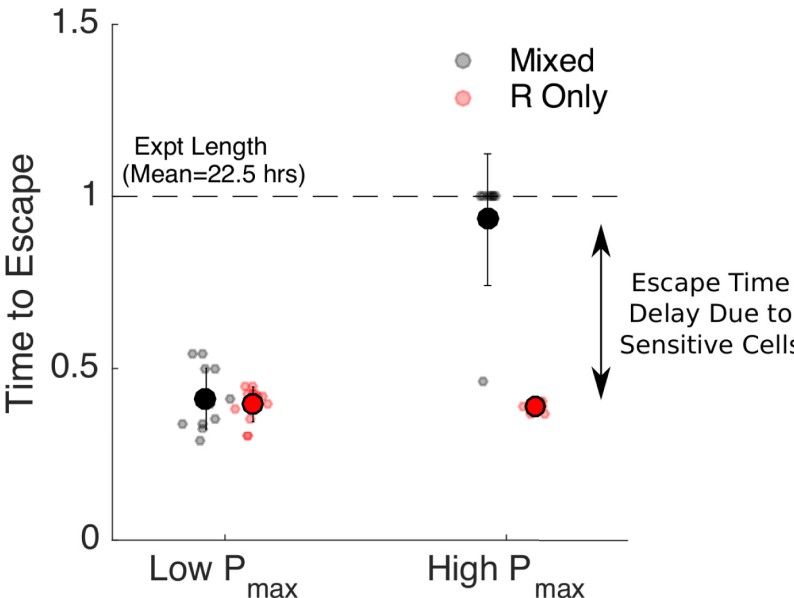

**Fig 4. Increased escape time under "competition maximization" requires a high threshold density.** Time to escape for populations maintained at low ($P_{max}$ = 0.1, left) and high ($P_{max}$ = 0.2, right) threshold densities ("acceptable burdens"). Small circles: escape times for individual experiments in mixed (black) or R-only (red) populations. Large circles: mean escape time across experiments, with error bars corresponding to ±1 standard deviation. Time to escape is defined as the time at which the population exceeds the threshold OD of $P_{max}$+0.025, in which the 0.025 is padding provided to account for noise fluctuations. Time to escape is normalized by the total length of the experiment (mean length 22.5 hours). Note that in the high $P_{max}$ case (right), the mixed population (black) reached the threshold density during the course of the experiment in only one case, so the escape times are set to 1 in all other cases. Data are deposited in the Dryad repository: https://doi.org/10.5061/dryad.s4mw6m943 [62]. OD, optical density; $P_{max}$, acceptable burden; R, resistant.

experiments at $P_{max}$ = 0.1 (Fig 3C). As before, the sensitive-only population is strongly inhibited by the drug and decreases in size over time (blue). Also as before, the resistant-only population (red) escapes the containment threshold, typically between 5 and 8 hours (faster than in the high $P_{max}$ experiment because of the lower threshold). In contrast to the previous experiment, however, the mixed population also escapes the containment threshold, and furthermore, it does so on similar timescales as the resistant-only population. This density-dependent discrepancy reflects the fact that sensitive and resistant populations interact when the density is sufficiently high. Again, the agreement between model and experiment is quite good, though the model does predict a slightly longer escape time in the mixed population. The small discrepancy between the model and the experiment suggests that at low densities the growth in the resistant-only vial is slower than the model predicts—suggesting that competition at low densities may be greater than the model assumes.

To quantify these results, we calculated the time to escape for each experiment. We defined time to escape for a particular experiment as the first time at which the growth curve (OD) exceeded the threshold density $P_{max}$ by at least 0.025 OD units (note that the 0.025 was chosen to allow for noise fluctuations in the OD time series without triggering a threshold crossing event). For low values of acceptable burden ($P_{max}$), the escape times for resistant-only and mixed populations are nearly identical (Fig 4, left). By contrast, at higher values of $P_{max}$, the escape time is dramatically increased (more than doubled) in the mixed population relative to the resistant-only population (Fig 4, right), even though both receive identical drug treatment and start with identically sized resistant populations. For our specific experimental setup, this corresponds to extending the escape time by more than 10 hours. Importantly, our

experiments suggest that sensitive cells are beneficial at high values of $P_{max}$ and have little effect at low $P_{max}$, consistent with the assumption that mutation-driven costs of sensitive cells in our system are negligible and that the acceptable burden must be high enough to derive benefit from competitive suppression.

To further quantify the relationship between $P_{max}$ and escape time, we performed similar experiments for a range of $P_{max}$ values (Fig 5). As predicted by the model, we found that escape time increases rapidly as $P_{max}$ increases for the mixed, but not for the resistant-only, population. Furthermore, the quantitative trends we observe are generally well captured by the model, though the theory consistently underestimates the escape time for the resistant-only vial (colored ×'s are above dashed curve) and the escape time at low $P_{max}$ values (blue ×'s and squares lie above both the dashed and dotted curve). This is consistent with the trend observed in Fig 3C and may suggest that the model underestimates the strength of competition at low densities. Finally, we performed one series of experiments over an extended time period (more than 35 hours, Fig 5C). To probe the extreme limits of containment, we chose $P_{max}$ to be greater than the predicted steady-state population size. In this scenario, both populations (resistant-only and mixed) eventually approach a population size below $P_{max}$. Interestingly, however, the resistant-only population transiently crosses $P_{max}$, whereas the mixed population can be held below the threshold for the entirety of the experiment. This example illustrates that there are situations in which populations containing resistant and sensitive cells can be held below threshold for extended periods, avoiding the transient escape dynamics of resistant-only populations.

## Discussion

In this work, we provide direct experimental evidence that the presence of drug-sensitive cells can lead to improved antibiotic-driven control of bacterial populations in vitro. Specifically, we show that a "competition-maximizing" strategy can contain mixed populations of sensitive and resistant cells below a threshold density for significantly longer than matched populations containing only resistant cells. The increase in escape time occurs only when the threshold density is sufficiently high that competition is significant. The findings are particularly remarkable given that experiments are performed in well-mixed bioreactors with continuous resource renewal, and even the highest density thresholds occur in the exponential growth regime for unperturbed populations. The surprisingly strong effect of competition under these conditions suggests that similar approaches may yield even more dramatic results in natural environments, in which spatial heterogeneity and limited diffusion may enhance competition [63–67].

Notably, our experiments did not uncover scenarios in which sensitive cells may actually be detrimental and accelerate the expansion of resistance. Theory suggests that these scenarios do indeed exist [45], but because of the typical mutation rates observed in bacteria, they cannot be reliably produced with our experimental system (see S2 Text for extended discussion). Indeed, the dose-response curves of isolates from the beginning of the experiment were similar to isolates taken a day later at the end of the experiment (S5 Fig), further supporting the idea that mutation plays a negligible role in this model. Although reservoir sizes and contamination risks limited the length of our experiments, it is possible that accumulated mutations over longer periods of time could change the average characteristics of the two populations. We note that we did observe similar dynamics (i.e., containment consistent with model predictions) in an isolated experiment run for more than 35 hours (Fig 5C).

Theory indicates that whether or not containment is better than elimination depends on a number of factors, including the frequency of resistance and how resistance impacts the basic growth (intrinsic fitness costs) and competitive ability (competitive fitness costs) of the resistant population (S6 Fig and [45]). These factors are important because they affect the relative

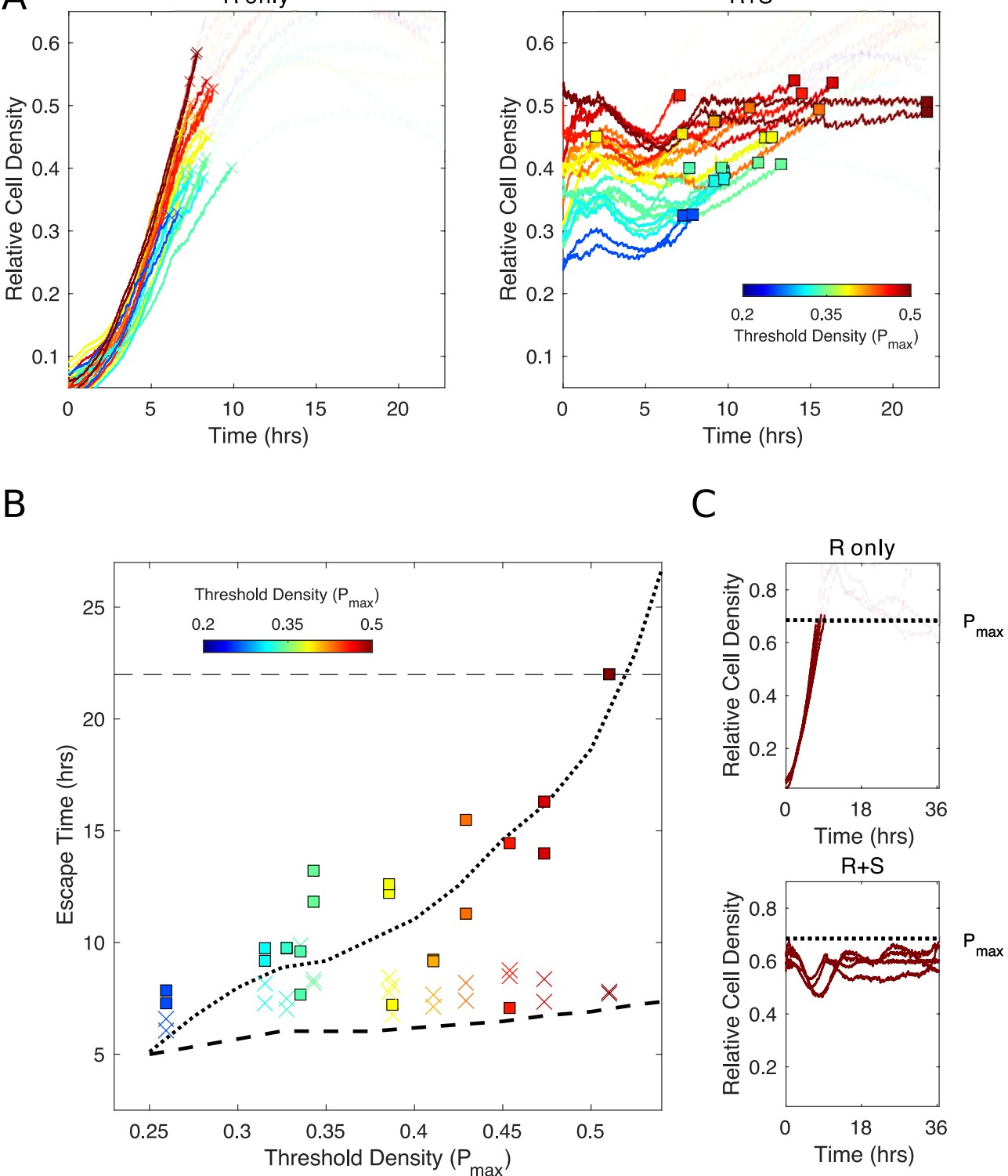

**Fig 5. Escape time depends sensitively on $P_{max}$ for mixed, but not for R-only, population.** (A) Growth curves for R-only (left) and mixed (right) populations for different values of $P_{max}$, ranging from 0.2 (blue) to 0.5 (red). Curves transition from opaque to transparent as they eclipse $P_{max}$, and the crossover point is marked with an "×" (R-only populations) or square (mixed populations). (B) Failure time (i.e., time to cross threshold) for experiments in panel A ("×" for R-only, squares for mixed populations; note that one yellow square falls below the axis limits). Dotted (dashed) curves are predictions from theoretical model for mixed (R-only) populations. (C) Example long-term (35+hours) experiment in the regime in which $P_{max}$ is larger than the steady-state population density. As in panel A, growth curves become transparent after crossing threshold. In all curves, density is

measured relative to carrying capacity in a unperturbed bioreactors (see S2 Fig). Data are deposited in the Dryad repository: https://doi.org/10.5061/dryad.s4mw6m943 [62]. $P_{max}$, acceptable burden; R, resistant; S, sensitive.

amounts of competition and mutational input. Here, we have fixed all of these factors and manipulated only the acceptable burden $P_{max}$, which we find must be sufficiently large to get significant benefit. In general, the threshold that is "high enough" will depend on the details of these other factors in potentially complex ways, and confirming these relationships experimentally is an exciting avenue for future work. Additionally, although in our experiment the mixed populations eventually became dominated by drug-resistant bacteria, in practice this may not always be the case. For certain combinations of fitness costs and competitive abilities, containment could drive the resistant population to very low levels. Importantly—even if this is not the case—our experimental results suggest that there are situations in which containment can extend the time before the resistant population dominates and treatment failure occurs.

It is important to keep in mind several technical limitations of our study. First, we measured population density using light scattering (OD), which is a widely used experimental surrogate for microbial population size but is sensitive to changes in cell shape [68]. Because we use protein synthesis inhibitors primarily at sub–minimum inhibitory concentrations, we do not anticipate significant artifacts from this limitation, though it may pose challenges when trying to extend these results to drugs such as fluoroquinolones, which are known to induce filamentation [69, 70]. In addition, in the absence of cell lysis, OD cannot distinguish between dead and living cells. However, our experiments include a slow background flow that adds fresh media and removes waste, leading to a clear distinction between nongrowing and growing populations. Under these conditions, fully inhibited (or dead) populations would experience a decrease in OD over time, whereas populations maintained at a constant density are required to divide at an effective rate equal to this background refresh rate.

Most importantly, our results are based entirely on in vitro experiments, which allow for precise environmental control and quantitative measurements but clearly lack important complexities of realistic in vivo and clinical scenarios. Developing drug protocols for clinical use is an extremely challenging problem. Our goal was not to design clinically realistic containment strategies but, instead, was to provide proof-of-principle experimental evidence that protocols aimed at maximally maintaining sensitive cells can contain resistance. The drug dosing protocol applied here attempts to supply the minimum possible amount of drug to control the population. However, the results show that it sometimes reduces the size of the sensitive population more than necessary, suggesting that our protocol is suboptimal, and therefore, our experimental results are likely underestimating the potential benefit of maximizing the sensitive population. The fact that we are still able to detect a benefit to maintaining a (slightly nonmaximal) sensitive population indicates that there is room for implementing these types of strategies in the nonidealized setting of real life. In fact, adaptive strategies designed to leverage (but not maximize) competition between drug-sensitive and drug-resistant cancer cells have been successful in both mouse models [42] and clinical cancer trials [46]. Our results suggest that these successes may be further improved by modifying treatment strategies to further increase competition—working toward competition-maximizing strategies.

We do not want to make a strong case that containment itself will soon become a widespread treatment strategy. The idea of a clinically acceptable pathogen burden will make many clinicians uneasy and, in many cases (e.g., bacterial meningitis), there is no acceptable burden. But in some settings, containment is not completely far-fetched. In acute infections, containment would be a temporary strategy, enough to relieve symptoms until immunity controls the infection [43]. Moreover, there is ample justification for the idea of an acceptable burden in

nonsterile sites (asymptomatic bacteriuria, gastrointestinal bacteria) and increasing evidence that a low burden of pathogen can be tolerated even in the lung or blood [71, 72]. Tolerance-promoting and antivirulence drugs are increasingly being sought [73–75], and these work by alleviating symptoms rather than pathogens. Additionally, there are settings in which adding drug-sensitive microbes is being considered (microbiome or bacteriotherapy [76]) or, in the case of fecal transplants, being enacted [77]. Moreover, in chronic infections with uncontrolled source populations, in which the evolution of resistance is the main threat to patient well-being [56], it is very possible that containment could be a key part of more complex resistance-controlling regimens. In cancer, in which resistance is responsible for many deaths, Gatenby and colleagues have argued that containment may better prolong life [31, 78, 79].

A basic science understanding of containment has the potential for real-world impact, as containment strategies are being tested on people now. If our hypothesis is confirmed, there are definable situations in which existing strategies could be improved by maximizing competition as opposed to simply using "less aggressive" adaptive approaches. We hope these experiments will motivate continued experimental, theoretical, and perhaps even clinical investigations, particularly in situations in which the primary threat to the well-being of the patient is resistance-induced treatment failure (e.g., [46, 56]).

## Methods

### Bacterial strains, media, and growth conditions

Experiments were performed with *Escherichia coli* strains REL606 and REL607 [59]. A resistant strain ("resistant mutant") was isolated from lab-evolved populations of REL606 undergoing daily dilutions (200×) into fresh media with increasing doxycycline (Research Products International) concentrations for 3 days. A single resistant isolate was used for all experiments. Stock solutions were frozen at −80˚C in 30% glycerol and streaked onto fresh agar plates (Davis Minimal Media [Sigma] with 2,000 g/ml glucose) as needed. Overnight cultures of resistant and sensitive cells for each experiment were grown from single colonies and then incubated in sterile Davis Minimal Media with 1,000 g/ml glucose liquid media overnight at 30˚C while rotating at 240 rpm. All bioreactor experiments were performed in a temperature-controlled warm room at 30˚C.

### Continuous culture bioreactors

Experiments were performed in custom-built, computer-controlled bioreactors as described in [58], which are based, in part, on similar designs from [57, 80]. Briefly, constant-volume bacterial cultures (17 mL) are grown in glass vials with customized Teflon tops that allow inflow and outflow of fluid via silicone tubing. Flow is managed by a series of computer-controlled peristaltic pumps—up to 6 per vial—which are connected to media and drug reservoirs and allow for precise control of various environmental conditions. Cell density is monitored by light scattering using infrared LED/detector pairs on the side of each vial holder. Voltage readings are converted to OD using a calibration curve based on separate readings with a table-top OD reader. Up to nine cultures can be grown simultaneously using a series of multiposition magnetic stirrers. The entire system is controlled by custom Matlab software.

### Experimental mixtures and setup

Before the experiments begin, vials are seeded with sensitive or resistant strains of *E. coli* and allowed to grow to the desired density in the bioreactor vials. Cells were then mixed (to create the desired population compositions) and diluted as appropriate to achieve the desired starting densities. Each vial is connected to (1) a drug reservoir containing media and doxycycline

(500 μg/ml), (2) a drug-free media reservoir that provides constant renewal of media, and (3) an effluent waste reservoir. Flow from reservoir 1 (drug reservoir) is determined in real time according to a simple feedback algorithm intended to maintain cells at a constant target density with minimal drug. Flow to/from reservoirs 2 and 3 provides a slow renewal of media and nutrients while maintaining a constant culture volume in each vial.

## Drug dosing protocols

To determine the appropriate antibiotic dosing strategy, the computer records the OD in each vial every 3 seconds. Every 3 minutes, the computer computes (1) the average OD, $OD_{avg}$, in the mixed vial over the last 30 seconds and (2) the current drug concentration in the vial. If $OD_{avg}$ is greater than $P_{max}$, the desired containment density, and the current drug concentration is less than $d_{max} = 125$ ng/mL ($d_{max} = 100$ ng/mL for the experiments used to generate Fig 5), then drug and media will be added to the vial for 21 seconds at a flow rate of 1 mL per minute. For the experiments corresponding to Fig 5, the drug threshold was lowered to 100 ng/mL to increase the long-term steady state of the system—allowing escape times for containment to be measured for a wider range of $P_{max}$ values. In a typical experiment, this control algorithm is applied to one of the mixed populations to determine, in real time, the drug dosing protocol (i.e., influx of drug solution over time). The exact same drug dosing protocol is then simultaneously applied to all experimental populations (mixed, resistant-only, sensitive-only, Fig 3). In parallel, an identical dosing protocol is applied to a series of control populations, but in these populations, the drug solution is replaced by drug-free media (S3 Text and S3 Fig). Finally, each experiment includes an unperturbed (no flow, no drug) control vial containing only resistant cells. To minimize effects of day-to-day fluctuations (in temperature, media batch, etc.), we measure population density in Fig 5 in units of carrying capacity estimated from these control populations each day.

## Whole-genome sequencing

Genomic DNA was isolated from single colony isolates using a Quick-DNA Fungal/Bacterial Kit (Zymo Reserach) according to the manufacturer's instructions. Libraries were prepared using the Swift 2S turbo flexible library prep kit and the swift normalase kit to normalize and pool libraries for sequencing. Sequencing was performed on the NovaSeq-6000, with 150-bp paired-end reads. Mutations were identified by mapping the Illumina sequencing reads to the reference strain REL606 (accession number NC_012967) using breseq [81].

Sequencing confirmed that the sensitive clone is identical to the published REL606 sequences (accession number NC_012967). The resistant strain has five mutations: OmpF P138H (CCT→CAT) porin mutation, a 1-bp deletion in fimbrial biogenesis outer membrane usher protein, an intergenic (+275/−54) C→T mutation between methylenetetrahydrofolate reductase/catalase/peroxidase HPI, an IS1 insertion into multidrug efflux transporter transcriptional repressor AcrR, and an IS150 mediated deletion of part of the ribose operon that has been previously described in the LTEE and confers a slight fitness advantage under those conditions [82]. Mutations in *OmpF* and *AcrR* genes have been previously linked with increased resistance to multiple antibiotics, including tetracyclines [60, 61].

## Mathematical model

The mathematical model used in the simulations is

$$\dot{S} = \frac{r_S}{1 + \left(\frac{D(t-\tau_S)}{h_S}\right)^{k_S}} \left(1 - \frac{S+R}{C}\right)S - \frac{F_D \chi_D + F_N}{V}S,$$

$$\dot{R} = \frac{r_R}{1 + \left(\frac{D(t-\tau_R)}{h_R}\right)^{k_R}} \left(1 - \frac{S+R}{C}\right) R - \frac{F_D \chi_D + F_N}{V} R,$$

$$\dot{R}_{only} = \frac{r_R}{1 + \left(\frac{D(t-\tau_R)}{h_R}\right)^{k_R}} \left(1 - \frac{R_{only}}{C}\right) R_{only} - \frac{F_D \chi_D + F_N}{V} R_{only},$$

$$\dot{S}_{only} = \frac{r_S}{1 + \left(\frac{D(t-\tau_S)}{h_S}\right)^{k_S}} \left(1 - \frac{S_{only}}{C}\right) S_{only} - \frac{F_D \chi_D + F_N}{V} S_{only},$$

$$\dot{D} = \frac{F_D \chi_D}{V} D_{in} - \frac{F_D \chi_D + F_N}{V} D, \tag{1}$$

where $S$ and $R$ are the drug-sensitive and drug-resistant densities in the mixed vial, $R_{only}$ is the bacterial density in the vial that contains only drug-resistant bacteria, $S_{only}$ is the bacterial density in the vial that contains only drug-sensitive bacteria, and $D$ is the drug concentration in the vials. The model parameters and initial conditions for the simulations (and experiments) are given in Tables 1–2. The effect of drug on growth rate is modeled as a Hill function with parameters $r_S$, $k_S$, and $h_S$ for the sensitive strain and parameters $r_R$, $k_R$, and $h_R$ for the resistant strain. There is also a time delay associated with the effect of drug (denoted by $\tau_S$ for the sensitive strain and $\tau_R$ for the resistant strain). Competition in the model is captured by using a logistic growth term with carrying capacity $C$. It is assumed that the sensitive and resistant strains have similar carrying capacities. Finally, the bioreactor has a continual efflux to maintain constant volume. The rate of this outflow is the sum of the constant background nutrient

**Table 1. Model parameter description.**

| Parameter | Definition | Value |
|---|---|---|
| $V$ | Volume of vial | 17 mL |
| $F_D$ | Flow rate of drug reservoir | $1\ \frac{mL}{min}$ |
| $\chi_D$ | Function indicating when drug is being added | 1 when drug is being added |
| $F_N$ | Constant background flow rate of nutrients | $0.067\ \frac{mL}{min}$ |
| $D_{in}$ | Drug concentration in drug reservoir | $500\ \frac{ng}{mL}$ |
| $C$ | Carrying capacity | Range: $(2.4 \times 10^8)$–$(3.2 \times 10^8)\ \frac{cells}{mL}$ (OD: 0.3–0.4)* |
| $r_S$ | Intrinsic per capita growth rate of drug-sensitive strain | $0.0169\ \frac{1}{min}$ |
| $r_R$ | Intrinsic per capita growth rate of drug-resistant strain | $0.0152\ \frac{1}{min}$ |
| $h_S$ | IC$_{50}$ for sensitive strain | $49.0639\ \frac{ng}{mL}$ |
| $h_R$ | IC$_{50}$ for resistant strain | $209.9995\ \frac{ng}{mL}$ |
| $k_S$ | Hill function coefficient | 2.2023 |
| $k_R$ | Hill function coefficient | 2.4849 |
| $\tau_S$ | Time delay for sensitive strain | 79.04 minutes |
| $\tau_R$ | Time delay for resistant strain | 96.72 minutes |

*Each experiment includes a drug-free control vial with no inflow or outflow.

Carrying capacity is estimated daily from this growth curve.

Abbreviation: IC$_{50}$, half-maximal inhibitory concentration; OD, optical density

**Table 2. Initial optical densities for simulations and experiments.**

|  | $P_{max}$ | $D(0)$ | $S(0)$ | $R(0)$ | $S_{only}(0)$ | $R_{only}(0)$ |
|---|---|---|---|---|---|---|
| High | 0.2 | 0 | 0.175 | 0.02 | 0.175 | 0.02 |
| Low | 0.1 | 0 | 0.075 | 0.02 | 0.075 | 0.02 |

Multiply optical densities by $8 \times 10^8$ to obtain cells per mL.

Abbreviation: $P_{max}$, acceptable burden

flow $F_N$ and any additional outflow required to compensate for the inflow of drug, which enters at a rate $F_D\chi_D$. $F_D$ is a constant rate, and $\chi_D$ is an indicator function, which is 1 when drug is being added to the vials and 0 when it is not. In the simulations, the decision of when to add drug is based on the same control algorithm that was used in the actual experiment (see Methods: Drug dosing protocols). Since the model describes the rate of change of bacterial density, the total efflux ($F_N+F_D\chi_D$) is divided by the volume of the vials $V$. The drug concentration in the vials is determined by the rate of drug flow into the vials ($F_D\chi_D D_{in}$, where $D_{in}$ is the concentration of drug in the reservoir) and the rate of efflux out of the vials ($F_N+F_D\chi_D$). The values of $D_{in}$, $V$, $F_D$, and $F_N$ were chosen to match the associated values in the experimental system; all other parameters in the model were fit using independent experimental data (see S3 Text for details) and are given in Table 1.

## Supporting information

**S1 Text. Details on parameter estimation.**
(PDF)

**S2 Text. Details on mutation.**
(PDF)

**S3 Text. Drug-free controls.**
(PDF)

**S1 Fig. Computer-controlled bioreactors.** Constant-volume bacterial cultures (17 mL) are grown in glass vials with customized Teflon tops that allow inflow and outflow of fluid via silicone tubing. Flow is managed by a series of computer-controlled peristaltic pumps that are connected to media and drug reservoirs. Cell density is monitored by light scattering using infrared LED/detector pairs on the side of each vial holder. Voltage readings are converted to OD using a calibration curve based on separate readings with a tabletop OD reader. Up to nine cultures can be grown simultaneously using a series of multiposition magnetic stirrers. The entire system is controlled by custom Matlab software. Flow chart (above) depicts adaptive drug therapy (lower branches) intended to maintain constant OD by adding drug in response to changes in cell density. LED, light-emitting diode; OD, optical density.
(PNG)

**S2 Fig. Growth of resistant cells in unperturbed bioreactors.** Cell density (OD) over time for REL607-derived resistant strains in bioreactors without influx or outflow of media. Transparent black lines correspond to growth curves performed in parallel with each bioreactor experiment. Thick black curve is the median over replicates. Dashed lines indicate threshold densities used in experiments ($P_{max} = 0.2$ and $P_{max} = 0.1$). Data are deposited in the Dryad repository: https://doi.org/10.5061/dryad.s4mw6m943 [62]. OD, optical density; $P_{max}$, acceptable burden.
(PDF)

**S3 Fig. Matched drug-free control populations are not contained by adaptive dosing proto-col.** Conditions are identical to those in Fig 3B and 3C except that all populations receive drug-free media rather than drug solution media as part of the adaptive dosing protocol. Data are deposited in the Dryad repository: https://doi.org/10.5061/dryad.s4mw6m943 [62].
(PDF)

**S4 Fig. Mixed populations contain primarily resistant cells at final time point of escape time experiment.** The REL606-derived resistant strain appears red, and the sensitive REL607 strain appears pink when grown on TA plates. Upper panels: samples from two mixed vials taken at the end of a high-density escape time experiment (as in Fig 3B). Arrows indicate sensitive colonies. Bottom row: samples from the end of a high-density escape time experiment for a vial seeded with only resistant bacteria (left) and a vial seeded with only sensitive bacteria (right). TA, tetrazolium arabinose.
(PDF)

**S5 Fig. Sensitive and resistant isolates show similar dose-response curves before and after experiment.** Dose-response curves measured in 96-well microplates for sensitive (REL607; blue circles) and REL606-derived resistant strains (red circles). Dark line, mean across replicates. Thin (transparent) curves correspond to colonies isolated from population mixture at the end of an escape time experiment. Red curve, resistant isolate (appears red on plate); blue curves, sensitive isolates (appear pink on plate). Data are deposited in the Dryad repository: https://doi.org/10.5061/dryad.s4mw6m943 [62].
(PDF)

**S6 Fig. Amount of benefit from containment depends on multiple factors.** Simulation showing fold increase in escape time gained by using containment instead of elimination. Each shaded region corresponds to a different $P_{max}$ (green, red, and blue are 20%, 30%, and 40% of the carrying capacity, respectively). Upper bounds of each shaded region correspond to an intrinsic fitness cost for resistance of 25% ($r_R = r_S(0.75)$), and lower bounds assume no fitness cost ($r_R = r_S$). Simulation uses mathematical model from main text and parameter values given in Table 1 (except for $r_R$, which is modified as described above). Trends show that increasing the intrinsic fitness cost and decreasing the frequency of resistance will increase the benefit of containment. Importantly, these simulations assume that there is no mutation. The role of fitness costs and frequency of resistance are more complicated when there is appreciable mutational input. Data are deposited in the Dryad repository: https://doi.org/10.5061/dryad.s4mw6m943 [62]. $P_{max}$, acceptable burden.
(JPG)

## Author Contributions

**Conceptualization:** Elsa Hansen, Jason Karslake, Robert J. Woods, Andrew F. Read, Kevin B. Wood.

**Formal analysis:** Elsa Hansen, Jason Karslake.

**Funding acquisition:** Robert J. Woods, Andrew F. Read, Kevin B. Wood.

**Investigation:** Elsa Hansen, Jason Karslake.

**Methodology:** Elsa Hansen, Jason Karslake, Robert J. Woods, Andrew F. Read, Kevin B. Wood.

**Supervision:** Robert J. Woods, Andrew F. Read, Kevin B. Wood.

**Validation:** Elsa Hansen, Jason Karslake, Robert J. Woods, Andrew F. Read, Kevin B. Wood.

**Visualization:** Elsa Hansen, Jason Karslake, Kevin B. Wood.

**Writing – original draft:** Elsa Hansen, Jason Karslake, Kevin B. Wood.

**Writing – review & editing:** Elsa Hansen, Jason Karslake, Robert J. Woods, Andrew F. Read, Kevin B. Wood.

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
