## [Editor Report · Decision Letter 0]

14 Jun 2019

Dear Dr Wood, 

Thank you for submitting your manuscript entitled "Antibiotics can be used to contain drug-resistant bacteria by maintaining sufficiently large sensitive populations" for consideration as a Research Article by PLOS Biology.

Your manuscript has now been evaluated by the PLOS Biology editorial staff, as well as by an academic editor with relevant expertise, and I'm writing to let you know that we would like to send your submission out for external peer review.

**Important**: Please also see below for further information regarding completing the MDAR reporting checklist. The checklist can be accessed here: https://plos.io/MDARChecklist

Please re-submit your manuscript and the checklist, within two working days, i.e. by Jun 18 2019 11:59PM.

Kind regards,

Roli Roberts

Senior Editor

PLOS Biology

INFORMATION REGARDING THE REPORTING CHECKLIST:

PLOS Biology is pleased to support the "minimum reporting standards in the life sciences" initiative (https://osf.io/preprints/metaarxiv/9sm4x/). This effort brings together a number of leading journals and reproducibility experts to develop minimum expectations for reporting information about Materials (including data and code), Design, Analysis and Reporting (MDAR) in published papers. We believe broad alignment on these standards will be to the benefit of authors, reviewers, journals and the wider research community and will help drive better practise in publishing reproducible research. 

We are therefore participating in a community pilot involving a small number of life science journals to test the MDAR checklist. The checklist is intended to help authors, reviewers and editors adopt and implement the minimum reporting framework. 

IMPORTANT: We have chosen your manuscript to participate in this trial. The relevant documents can be located here:

MDAR reporting checklist (to be filled in by you): https://plos.io/MDARChecklist

**We strongly encourage you to complete the MDAR reporting checklist and return it to us with your full submission, as described above. We would also be very grateful if you could complete this author survey:

https://forms.gle/seEgCrDtM6GLKFGQA

Additional background information:

Interpreting the MDAR Framework: https://plos.io/MDARFramework

Please note that your completed checklist and survey will be shared with the minimum reporting standards working group. However, the working group will not be provided with access to the manuscript or any other confidential information including author identities, manuscript titles or abstracts. Feedback from this process will be used to consider next steps, which might include revisions to the content of the checklist. Data and materials from this initial trial will be publicly shared in September 2019. Data will only be provided in aggregate form and will not be parsed by individual article or by journal, so as to respect the confidentiality of responses. 

Please treat the checklist and elaboration as confidential as public release is planned for September 2019.

We would be grateful for any feedback you may have.

---

## [Decision Letter · Decision Letter 1]

28 Jul 2019

Dear Dr Wood,

Thank you very much for submitting your manuscript "Antibiotics can be used to contain drug-resistant bacteria by maintaining sufficiently large sensitive populations" for consideration as a Research Article at PLOS Biology. Your manuscript has been evaluated by the PLOS Biology editors, an Academic Editor with relevant expertise, and by three independent reviewers.

The reviews of your manuscript are appended below. You will see that the reviewers find the work potentially interesting. However, based on their specific comments and following discussion with the academic editor, I regret that we cannot accept the current version of the manuscript for publication. We remain interested in your study and we would be willing to consider resubmission of a comprehensively revised version that thoroughly addresses all the reviewers' comments. We cannot make any decision about publication until we have seen the revised manuscript and your response to the reviewers' comments. Your revised manuscript would be sent for further evaluation by the reviewers.

We appreciate that these requests represent a great deal of extra work, and we are willing to relax our standard revision time to allow you six months to revise your manuscript. Please email us (plosbiology@plos.org) to discuss this if you have any questions or concerns, or think that you would need longer than this. At this stage, your manuscript remains formally under active consideration at our journal; please notify us by email if you do not wish to submit a revision and instead wish to pursue publication elsewhere, so that we may end consideration of the manuscript at PLOS Biology.

Your revisions should address the specific points made by each reviewer. Please submit a file detailing your responses to the editorial requests and a point-by-point response to all of the reviewers' comments that indicates the changes you have made to the manuscript. In addition to a clean copy of the manuscript, please upload a 'track-changes' version of your manuscript that specifies the edits made. This should be uploaded as a "Related" file type. You should also cite any additional relevant literature that has been published since the original submission and mention any additional citations in your response. 

Before you revise your manuscript, please review the following PLOS policy and formatting requirements checklist PDF: http://journals.plos.org/plosbiology/s/file?id=9411/plos-biology-formatting-checklist.pdf. It is helpful if you format your revision according to our requirements - should your paper subsequently be accepted, this will save time at the acceptance stage.

Please note that as a condition of publication PLOS' data policy (http://journals.plos.org/plosbiology/s/data-availability) requires that you make available all data used to draw the conclusions arrived at in your manuscript. If you have not already done so, you must include any data used in your manuscript either in appropriate repositories, within the body of the manuscript, or as supporting information (N.B. this includes any numerical values that were used to generate graphs, histograms etc.). For an example see here: http://www.plosbiology.org/article/info%3Adoi%2F10.1371%2Fjournal.pbio.1001908#s5.

For manuscripts submitted on or after 1st July 2019, we require the original, uncropped and minimally adjusted images supporting all blot and gel results reported in an article's figures or Supporting Information files. We will require these files before a manuscript can be accepted so please prepare them now, if you have not already uploaded them. Please carefully read our guidelines for how to prepare and upload this data: https://journals.plos.org/plosbiology/s/figures#loc-blot-and-gel-reporting-requirements.

Upon resubmission, the editors will assess your revision and if the editors and Academic Editor feel that the revised manuscript remains appropriate for the journal, we will send the manuscript for re-review. We aim to consult the same Academic Editor and reviewers for revised manuscripts but may consult others if needed.

If you still intend to submit a revised version of your manuscript, please go to https://www.editorialmanager.com/pbiology/ and log in as an Author. Click the link labelled 'Submissions Needing Revision' where you will find your submission record. 

Sincerely,

Roli Roberts

Senior Editor

PLOS Biology

REVIEWERS' COMMENTS:

Reviewer #1:

The authors develop a computer controled bioreactor experimental model to assess whether a strategy of ‘containment’ (drug treatment only to limit growth below a threshold) can increase time to escape from drug control. They provide exciting experimental support for the containment strategy, illustrating a significant delay to population growth under a containment regime (compared to a limit case of total S annihilation) – even when the containment target is substantially below the bacterial carrying capacity. 

These results present an important avenue for future development. My two main issues with the current MS are related and fixable primarily in the discussion – first, how could a containment strategy be implemented in the clinic? Second, how does a containment strategy relate to other conditional strategies and to an unconditional low dose strategy? 

1) implementation. 

The experimental feedback control mechanism doesn’t appear to be a plausible clinical strategy (and I’m fine with the proof of concept goals) – but this leaves the key question of translation. What do the authors see as a path to capturing elements of this feedback control in the clinic? It might be interesting at this point to borrow from analogous approaches in the cancer adaptive therapy field, which I believe is more developed. Does the approach work, given uncertainty over the R/S composition of the infection? How can clinicians track infection growth? Imaging? Symptoms?? 

2) relationship to other conditional/unconditional strategies

A full clinical implementation of this experimental model would require information on the initial resistance distributions and also the temporal dynamics of an infection – this is a tough goal (see point 1). However just having resistance profiles alone could open simple and effective treatments – use a drug that is effective against both co-infecting strains. (https://journals.plos.org/plosbiology/article?id=10.1371/journal.pbio.3000250). In light of this simpler strategy (treat conditional on resistances), why invest in containment? One argument is pan-resistant strains – conditioning on resistance is no longer effective, and conditioning on pathogen growth might buy time for … immune control? It would also be useful to discuss relationships with the most simple and low tech strategy of reducing drug dose – which will tend to reduce competitive release. This would be an interesting avenue for a control experiment / simulation (same total drug, constant administration) 

other concerns:

Fig 4: please use time in hours on y axis. The timescale here is important in my view to discuss. You’re potentially adding hours to the control window – not a substantial time period. This ties into another key translational theme – what is the clinical advantage of adding hours to the window of time until loss of infection control? In an acute infection context this might be of practical significance for development of adaptive immune control – not sure here, would be interested in authors thoughts. 

Related to timescale issue - what were the obstacles to running the experiment for longer? It would have been useful to compare the ultimate escape outcomes in all treatments and replicates. More data on longer timecourses would be helpful- the authors should at least comment on what the road block was to longer implementation of this device. 

Minor: 

How resistant is the ‘R’ E. coli strain? Sounds like a low ½ concentration. Not a big concern as this is plainly a ‘proof of concept’ implementation.

Reviewer #2:

In the light of the worldwide spread of multi-drug resistant bacterial pathogens, it is important to develop new approaches that limit the spread of superbugs and make current antibiotic therapies more sustainable. The paper by Hansen et al. contributes to this aim and asks whether less aggressive antibiotic treatments, allowing competition between susceptible and resistant strains could repress overall population growth, thereby keeping pathogen burden below an acceptable level and extend treatment efficacy.

I concur with the authors that inter-strain competition could be a powerful way to limit the spread of antibiotic resistant strains. But unfortunately the experimental design and the data presented are quite weak, and I am not convinced that this study will extend our knowledge on the role of competition in making antibiotic therapies more sustainable.

The authors performed bioreactor (chemostat) experiments with Escherichia coli bacteria. They have three treatments: (1) a monoculture of a susceptible strain, (2) a monoculture of an antibiotic resistant strain, and (3) a mixture of the two. Through the automated adjustment of the flow in the bioreactor, population growth rate can be kept constant under a range of antibiotic concentrations. The authors chose two arbitrary “acceptable” pathogen levels (OD = 0.225 and OD = 0.125), and measured the time for the treatments needed to surpass this threshold. The time needed to do so is called “escape time” and is interpreted as time until treatment failure. The authors found that (i) the escape time was very short for the antibiotic resistant strain, (ii) sensitive populations declined under treatment and never surpassed the threshold, and (iii) mixed populations had either an extended escape time or did not surpass the threshold. The authors conclude that competition helps to keep bacterial load at a manageable level. While I have no problem with these results, I question the overall relevance of the finding.

(1) The authors consider treatment 2 (resistant bacteria only) as a “mutation minimization” treatment, matching aggressive antibiotic treatment. However, aggressive treatment is thought to minimize the risk of resistance to arise in susceptible populations and not in already resistant populations as implemented by the authors. In this context, it is not surprising that resistant monocultures grow best. Anything else would be surprising, and this treatment does not match a “mutation minimization” treatment as one would deploy in clinical settings.

(2) The authors consider treatment 3 as a “competition maximization” treatment, and conclude that competition is only efficient in extending escape time at higher population densities. I have multiple problems with this conclusion. (a) Higher bacterial burden is likely bad for the patient and this is not a practicable way to take. (b) the chosen thresholds are arbitrary and the growth increase observed for the mixed treatment under low density (Fig. 3C) is minimal. It surpasses the threshold, yes, but compared to the resistant only treatment, mixed cultures are still much more stable and show a dramatically lower growth increase. Thus, competition also has an effect at low cell density. (c) the observed effects are transient. The authors plated the mixed cultures at the end of their experiment and found that populations are dominated by resistant clones. This means that in the long-run competition cannot constrain the spread of resistance.

(3) The advantage of in vitro proof-of-concept studies is that there is full control over the experimental set up and a large number of conditions and treatment combinations can be tested. Unfortunately, the authors did not make use of this advantage and the data presented is thin both in terms of quality and quantity. (a) The authors use an undefined laboratory evolved antibiotic resistant mutant. We don’t know the resistant mechanism involved and the potential mechanisms of competition that could drive interaction patterns in mixed cultures. (b) Only one antibiotic was used for hypothesis testing. So we don’t learn whether the observed patterns are of general relevance or specific to this antibiotic. (c) Only two populations densities were examined and we thus don’t learn what the overall relationship between escape time and population density is. This is particularly problematic because OD = 0.2 is called high density although in absolute terms OD = 0.2 is still very low density. From a proof-of-concept study, which is far away from anything that is clinically relevant, I would expect a much larger parameter space to be explored, especially with the mathematical model at hand.

More minor comments:

- It is unclear what should be extracted from the left panels of Figure 2. Is this just to demonstrate that real-time growth rates can be measured in the chemostats? This is quite trivial because this is what chemostats are made for.

- Why does the resistant strain treatment start with a much lower OD than the other two treatments? This induces a massive bias towards longer escape times for this treatment.

- page 2, line 3: “aggressive treatment may actually promote the emergence of resistance” sounds like the treatment itself will actively give raise to resistance mutation. Consider rephrasing to “promote selection for resistance” or “promote the spread of resistance”

- end of page 2 and beginning of page 3: I don’t think that statements like “this is the first explicit demonstration” and “we are also unaware of any direct tests …” really help to increase the impact/relevance of the paper. Please remove.

- page 3: “the celebrated long term evolution experiment” reads odd. Please be more objective.

Reviewer #3:

The aim of this study is to provide a proof-of-concept experimental illustration that under given conditions, containment (i.e. more moderate) drug regimens can better manage antibiotic resistance than elimination (i.e. aggressive) strategies. For this the authors design three scenarios in an in-vitro experimental setup with automated drug and nutrient delivery to study the effect of treatment strategy on resistance emergence: i) resistant-only (no competition), ii) resistant+sensitive cells (competition scenario) and iii) sensitive-only dynamics. They set the drug concentration adaptively in the mixed scenario, to maintain the population at a certain density, and apply the same drug concentration to the other homogeneous populations. The readout for time of escape is the time it takes each population to cross a given target threshold density. The study finds that in the mixed case, the time of escape is higher than in the resistant only case, thus indicating the important role of competition in delaying resistance selection in real settings. This finding is also corroborated by a mathematical model. The authors go on to test this effect at another population size, and report that at lower population sizes the benefit of competition in reducing resistance is lower. Overall, I consider the study interesting and novel in the combination of ecological theory, experimental in vitro-approach with computer-controlled bioreactors and mathematical modeling. Indeed, the question of optimal drug therapies is a very urgent one in the face of antibiotic resistance. Previous mathematical models using population dynamic arguments have shown that competition between drug-sensitive and drug-resistant pathogens can slow down the selection of resistance. This particular study goes one step further by providing experimental validation of this expectation. However, there are some parts of the study and analysis that I find confusing and that I would need the authors to clarify. 

1. I am a bit confused by the authors' focus and repeated reference to "emergence of resistance" and "escape time", since they do not talk explicitly about the mechanisms underlying this escape. Is it because of selection of pre-existing sub-populations, or because of further or de-novo mutations? In particular, are the mechanisms of "escape" the same in the Resistant-only, mixed case, and Sensitive-only scenarios? In general I guess not, and I would invite the authors to comment a bit more on this biological aspect. If I understand it correctly, in the Resistant-only scenario, the process governing escape is basically growth of resistant cells under sub-inhibitory antibiotics and the time it takes to exceed a certain population size threshold. In the mixed case, escape is basically due to the selection of the resistant sub-population in the context of a heterogeneous population, where the majority of sensitive cells by competitive suppression, delay any growth advantage of the resistant minority in the presence of the drug. In the sensitive-only case, escape can only happen if there is an allegedly very rare beneficial mutation conferring drug resistance in the time frame of the experiment followed by subsequent growth. Thus, there should be no emergence per se in this study, only selection. This is also confirmed by the dose-response curve for R-only population at the end of the experiment, which is the same as the original strain.

2. Another issue the authors seem to overlook is the fact that mutational processes can happen both in the sensitive cells and already resistant cells. There are many studies that show that drug-resistant cells may experience even higher mutation rates that effectively accelerate further resistance evolution. In the present study, this process is completely neglected. Why? 

3. Regarding, the definition and approach to competition adopted in this study, I would also invite the authors to elaborate more and be more explicit on: 

i) the confounding between population size and heterogeneity in their 3 scenarios: Besides the fact that we are dealing with a homogeneous vs. heterogeneous population, there is also a big difference in initial population size in the R-only (much lower than Pmax) and mixed case (equal to Pmax). This by default creates an advantage for the population of cells growing in the R-only regime. And this advantage is higher, the more space there is for growth from P(0) to Pmax. Thus, in one case we are dealing with a growing ('epidemic scenario') population, in the other case, we are dealing with an 'endemic' scenario, of constant population size and simple replacement. It is known also from epidemiological models and studies that patterns of evolution can be different. 

ii) What is the rationale behind equalizing the drug concentrations across the scenarios? By equalizing drug concentrations (set by the mixed regime) the authors are comparing dynamics using minimal inhibitory treatment vs. sub-inhibitory treatment relatively in the "mixed/sensitive-only" and R-only scenarios. Thus, unlike the schematic shown in Figure 1A, the resistant-only scenario, is experiencing much lower concentration of drug than its corresponding aggressive regime, amplifying the total growth advantage relative to the mixed case. Thus, not only they are starting with dR/dt>0, not only resistant cells are not competing with sensitive cells, they are also under less drug-inhibition overall. It seems that the authors are simulating in the R-only regime a treatment of the type "assumed prior aggressive treatment that removed the S population + moderate treatment when there is only the resistant population left". 

iii) Competition maximization to slow down resistance is one thing, infection clearance is another. What are the novel prospects for infection clearance generated by this study? In light of this work, could addition of drug-sensitive cells to a drug-resistant infection be used to increase competition and leverage its benefits? 

4. It is necessary for this study to address more clearly what the effect of containment strategies is along the 2-d continuum: pop. size vs. initial frequency of R. The authors explore indeed two population sizes (achieved with different drug dosing), but these correspond at the same time to two different initial frequencies of resistance in the bacterial population, namely 10% in the high population size and 20% in the small population size. Thus, we cannot really disentangle whether the sensitivity of escape time is a population size effect or frequency-dependent effect, or of both. The higher the initial frequency of resistance in the mixed case (i.e. here the lower the Pmax), the closer we are to the R-only extreme, and that is why no difference is found in "escape time." I recommend the authors to comment on this issue when interpreting their results.

In my view, a contour plot of the model-derived expected time to escape as a function of both population size and initial frequency would be an important and insightful addition to the paper. I would even suggest to provide two versions of such contour plot, for example by changing the growth differential between the sensitive and resistant strains or efficacy of antibiotic. In this way, it will become clear that the quantitative benefit of containment does not depend just on population size (as suggested in the paper and abstract), but is indeed much more complex, and may vary also with all these other parameters and initial conditions. 

5. The worst-case scenario for containment is when the total bacterial population is resistant (frequency of resistance=1) and when the population size is large. This scenario cannot be tested with the current framework because the maximum dose feasible is well below the half-maximum inhibitory concentration for the resistant population (129<209). Can the authors comment on this scenario? What would be the role of competition in that case? 

Since this study assumes that sensitive and resistant cells compete equally, can we expect a positive effect of competition also in the R-only adaptive regime? Would a moderate regime delay further resistance evolution compared to an aggressive regime? 

6. Some more links with existing models from the literature are warranted, e.g. Colijn and Cohen (eLife 2015, https://doi.org/10.7554/eLife.10559.001) where competition between a drug-sensitive strain and drug-resistant strain is modeled in more detail. Similarly, Gjini & Brito (Plos Comp. Biol. 2016 https://doi.org/10.1371/journal.pcbi.1004857) also consider in detail adaptive treatment, and along the lines of this study, conclude that at higher symptom thresholds, where indirect competition via immune stimulation is maximized, resistance selection risk is also lower, as opposed to lower symptom thresholds. This may be relevant for in vivo settings where antigen-dependent immunity acts as another layer of competition on heterogeneous bacterial populations. 

Similarly, some discussion on the possibility of asymmetric competition is also needed. Here it is assumed that the carrying capacities are the same for both S and R. Indeed, if the competition were asymmetric and S could inhibit S more or less than inhibiting R, and so on (there are in principle 4 pairwise competitive interactions), additional indirect effects could manifest on R dynamics due to S removal by the drug. This phenomenon has been discussed widely in pneumococcus epidemiology and serotype replacement due to vaccination, where asymmetric competition can accelerate or slow down non-vaccine type 'selection' (similar to R selection in the mixed experimental setting of this study).

Minor comments:

- Why is the main result in Figure 4 not divided in Probability of Escape and Time of Escape | Escape?

- In Table 2, the R-only and S-only initial conditions have to be corrected. Please also put the initial conditions in terms of bacterial concentrations as required by the differential equation model.

- More details on the Model fitting and parameter estimation, especially the delays associated with drug effect on resistant strain and sensitive strain are needed. How important are the assumed differences in these delays?

-Figure 3 should be accompanied in the Supplements with the corresponding underlying dynamics of replacement of S by R in the mixed regime during drug administration. The R-dynamics in that case could be compared with the R-only dynamics by overlaying the two graphs.

- In Figure 3, please clarify what generates the variability in model predictions. What is the stochastic component of the simulations?

---

## [Decision Letter · Decision Letter 2]

20 Mar 2020

Dear Dr Wood,

Thank you for submitting your revised Research Article entitled "Antibiotics can be used to contain drug-resistant bacteria by maintaining sufficiently large sensitive populations" for publication in PLOS Biology. I have now obtained advice from two of the original reviewers and have discussed their comments with the Academic Editor. 

Based on the reviews, we will probably accept this manuscript for publication, assuming that you will modify the manuscript to address the remaining points raised by the reviewers. Please also make sure to address the data and other policy-related requests noted at the end of this email.

Specifically, please address the remaining requests from reviewer #3 regarding the Supplementary Figs, and attend to my Data Policy request below.

We expect to receive your revised manuscript within two weeks. Your revisions should address the specific points made by each reviewer. In addition to the remaining revisions and before we will be able to formally accept your manuscript and consider it "in press", we also need to ensure that your article conforms to our guidelines. A member of our team will be in touch shortly with a set of requests. As we can't proceed until these requirements are met, your swift response will help prevent delays to publication.

*Copyediting*

*Published Peer Review History*

*Early Version*

*Submitting Your Revision*

Sincerely,

Roli Roberts

Senior Editor

PLOS Biology

DATA POLICY:

Regardless of the method selected (we note that you have expressed an intention to deposit your data in Dryad), please ensure that you provide the individual numerical values that underlie the summary data displayed in the following figure panels as they are essential for readers to assess your analysis and to reproduce it: Figs 2, 3, 4, 5, S2, S3, S4, S6, NOTE: the numerical data provided should include all replicates AND the way in which the plotted mean and errors were derived (it should not present only the mean/average values).

REVIEWERS' COMMENTS:

Reviewer #1:

The authors have made valuable edits in response to my earlier comments. I have no further concerns. 

minor: fix typo in title of figure 5 (which is a very nice addition)

Reviewer #3:

I appreciate the effort gone into this revision and the clarifications made by the authors to address most of the points raised. I do believe this has improved substantially the manuscript. While still some caveats remain, the authors have mentioned them more transparently in this version, opening the space for future work to fill the gaps. Overall, this is a valuable contribution to the empirical understanding of competition and intervention processes on the dynamics of bacterial populations, which may help address therapeutic challenges for resistant infections in clinical settings. 

Some final comments below: 

- As a further clarification on Figure 3, the provided Figure S4 is not what I had in mind when I asked for a comparison between Resistant dynamics visualization in the R-only vs mixed case. In its current version, I don't find figure S4 informative. I understand the explicit R-dynamics in the mixed case may be inaccessible in retrospect, thus making my request unfeasible with the current data, where only total population size is tracked over time. I would recommend to remove figure S4. 

- I would appreciate seeing the figure about Frequency of Resistance and escape time in the response file (response to my previous comment 4) as an added Figure to the Supplements, with appropriate referencing in the main text.

- I suggest removing unnecessary brackets from the model equations (e.g. from the numerator in some fractions), this would make the reading of the model less cumbersome.

---

## [Editor Report · Decision Letter 3]

23 Apr 2020

Dear Dr Wood,

On behalf of my colleagues and the Academic Editor, David S. Schneider, I am pleased to inform you that we will be delighted to publish your Research Article in PLOS Biology. 

Early Version

PRESS 

Kind regards,

Vita Usova 

Publication Assistant, 

PLOS Biology

on behalf of

Roland Roberts,

Senior Editor

PLOS Biology